# Analysis of Perceptions, Preferences, and Participation Intention of Urban Forest Healing Program among Cancer Survivors

**DOI:** 10.3390/ijerph20021604

**Published:** 2023-01-16

**Authors:** Eun Young Park, Min Kyung Song, Sang Yi Baek

**Affiliations:** 1College of Nursing, Gachon University, Incheon 21936, Republic of Korea; 2Department of Nursing, College of Medicine, University of Ulsan, Ulsan 44610, Republic of Korea

**Keywords:** cancer survivors, urban forest healing, survey

## Abstract

It is important to manage the health of cancer survivors who have returned to their daily lives. An increasing number of cancer survivors are undertaking health care in forests near their residences. This cross-sectional study aimed to determine the perceptions of forest healing and the program preferences of adult cancer survivors. Data were collected from 388 female cancer survivors through an online survey. Breast cancer survivors comprised 66.2%, and 63.6% of the study participants were diagnosed with cancer within 2 to 5 years prior to the study. The subjective health status was 2.68 ± 0.61 out of 4 points. Of the participants, 62.1% had heard of forest healing and 65.7% intended to participate in the forest healing program. Participants who frequently visited the forest were more likely to participate in the forest healing program in the future than those who rarely visited the forest. A survey among 255 people on specific preferences for the program found that the regular visit type was preferred over the one-time visit type. These results are meaningful because they can be used as a basis for the development of a forest treatment program that reflects the needs of cancer survivors.

## 1. Introduction

Cancer survivors are defined as those who are alive after a cancer diagnosis [1]. Advances in the early detection and treatment of cancer have significantly increased the number of cancer survivors worldwide. In the United States, more than 16.9 million people were diagnosed with cancer in 2019 [2]. The cancer prevalence in Korea was 4.2% in 2019, and the chance of developing cancer where people live to the life expectancy age reached 37.9% [2]. Approximately 50% of patients diagnosed with cancer have survived for more than 10 years [2,3,4]. Thus, the health problems of cancer survivors who survived after being diagnosed with cancer have been considered a crucial public health problem in many countries [5].

Many people who have undergone cancer treatment experience physical problems related to cancer treatment and are at risk of developing long-term side effects, such as secondary cancer, comorbidities, and psychosocial problems [6]. These side effects can occur months or years after treatment. Evaluating and treating late effects is an important aspect of cancer survivorship care [7,8]. In the low-risk survivor group, health management after treatment is similar to the general public’s health promotion activities regarding diet and exercise for proper weight management and nutritional status [7,9].

Among female cancer survivors, those with breast cancer experience numerous health problems after diagnosis. Specifically, fatigue, sleep problems, osteoporosis, and lymphedema significantly increase after treatment [10]. However, many cancer survivors are unable to implement healthy lifestyle practices to improve their lifestyle habits for many reasons, including the long-term effects of cancer treatment [9]. In particular, female cancer survivors have lower subjective health statuses than males, higher stress levels than males, and a lower quality of life due to higher levels of depression and anxiety [11,12]. In addition, in Korea, it is difficult for female cancer survivors to receive appropriate support at home due to the stereotype of gender roles at home [13].

Nature’s beneficial influence appears to extend beyond individuals. Accumulating evidence supports a positive relationship between exposure to natural environments and other-oriented prosocial behavior. Thus, when health deteriorates, people turn to nature for recovery. In health management strategies for people with cancer, natural activities such as walking in the forest can have healing effects such as stress reduction, immunity enhancement, and anticancer effects [14,15]; these activities have recently been conducted in forests for health promotion and disease treatment [16,17]. In addition to the physiological effects on patients and survivors of cancer, including increased immunity, reduced stress, and anticancer effects [9,10], activities in the forest have a positive effect on emotional stability and self-actualization and help relieve depression [18,19,20]. Most of the programs with proven effectiveness are stay-type programs in forests far from cities [14,15,16]. However, since most cancer survivors are people who lead a normal life, it is more convenient to access a program in which they regularly visit the forest near their residence rather than a stay-type program.

Identifying potential subjects’ needs in developing the forest healing program is important [21]. It is necessary to investigate the perception, preferences, intentions, and motivation of participants in developing a forest healing program for cancer survivors. The intention to participate is considered a crucial factor for predicting the participation rate in a program [22]. Therefore, identifying factors that affect the intention to participate when developing the forest healing program might lead to positive results in improving the actual participation rate.

Although some studies have focused on the needs, preferences, and participation intentions of general adults and chronically ill patients regarding forest healing programs [23,24,25], to our knowledge, no previous study has investigated the needs, preferences, and participation intentions of cancer survivors.

Therefore, this study aims to identify the perception, preference, and intention to participate in a forest healing program targeting female cancer survivors who are potential consumers and to identify the factors that affect the intention to participate in the program. This study will form the basis for developing and establishing a forest healing program for cancer survivors in urban forests in the future.

## 2. Materials and Methods

### 2.1. Study Design and Participants

This cross-sectional study aimed to determine the perceptions of forest healing and the program preferences of adult cancer survivors. The participants of this study were female adult cancer survivors aged 20 years or older who were within 5 years of being diagnosed with cancer and were completing active treatment (surgery, standard chemotherapy, and radiation therapy). Participants were recruited between May and June 2021. To recruit participants, leaflets were posted on the social media of the Cancer Survivor Integrated Support Center (National Cancer Center, Incheon, Gyeonggi, and Busan) and the Cancer Education Center of two hospitals in Seoul.

### 2.2. Ethical Approval

This study was conducted with ethical approval from the institutional review board. The purpose, method, and time required for the study were explained to the participants, and they voluntarily participated in the study. They were informed that they could withdraw their participation at any time. To maintain anonymity, the collected data were coded so that the participants’ information could not be identified, and their personal information was kept confidential by storing it in the form of a password-protected file. This study was conducted per the Declaration of Helsinki.

### 2.3. Measures and Data Collection

#### 2.3.1. Questionnaire

The survey was created using Google Forms and consisted of three parts. The first part included seven questions about demographic characteristics. The second part comprised five items on clinical characteristics. Finally, the third part referred to forest healing and included awareness of forest healing, experience participating in forest healing programs, purpose of participating in forest healing, and preferred types and components of forest healing programs. It was mandatory to respond to all items of the online survey to ensure that there were no missing survey items.

The validity of the questionnaire was evaluated by an expert group consisting of a nursing professor, a forest healing instructor, and a researcher from the National Institute of Forest Science. The item validity index was calculated based on the relevance rating given by experts, and all items were 0.80 or higher.

#### 2.3.2. Data Collection

Data were collected using Google Forms between 25 May and 10 June 2021. The first screen of the online survey provided information about the study (i.e., purpose, procedure, benefits and risks, and protection of the rights of the participants). Informed consent was obtained prior to starting the survey. The time taken to answer the questionnaire was approximately 20 min. The calculated sample size was 305 when effect size = 0.25 and power = 0.95. A total of 458 people who understood the study’s purpose and agreed to participate in the online survey participated. Among them, 6 submitted duplicate questionnaires, 33 did not meet the inclusion criteria (i.e., currently under treatment or more than five years after the end of treatment), 5 submitted incomplete questionnaires, and 26 were male responders. After excluding the above, 388 responses were analyzed.

### 2.4. Statistical Analyses

Statistical analyses were performed using SPSS (version 21.0) software, and the significance level was set at *p* < 0.05. The participants’ demographic and clinical characteristics, interest in forest healing, and preference for forest healing programs were expressed using frequency and percentages, mean, and standard deviation. Frequency and multiple response analyses were conducted for data analysis. A normality test was performed to determine the degree of difference in the purpose of participating in the forest healing program by cancer types, but it was not normally distributed; therefore, the Kruskal–Wallis test, a nonparametric test, was performed. The relationship between cancer survivors’ intention to participate in forest healing programs and the demographic and clinical variables was investigated using the binary logistic regression model. The independent variables were age, educational level, marriage status, caregivers, employment status, monthly household income, place of residence, cancer type, end of treatment period, subjective health status, presence of other diseases, awareness of forest therapy, forest visit frequency, and forest healing experience. The statistical significance level was based on *p* < 0.05.

## 3. Results

### 3.1. Participant Characteristics

#### 3.1.1. Demographic Characteristics

The age of the participants was 45.84 ± 7.96, and 52.6% of the participants were in their 40s. In terms of marital status, 72.7% were married. The main nursing providers were spouses and children (50.8%) and self-care (33.8%). Regarding employment status, 67.3% were unemployed, and their monthly household income was between KRW 2 to 5 million (42.5%) and KRW 5 million or more (42.3%). Of the participants, 73.5% lived in the capital city. (Table 1).

#### 3.1.2. Clinical Characteristics

The cancer types included breast (66.2%), gynecologic (11.1%), thyroid (8.2%), gastrointestinal (7.7%), and other cancers (6.7%). The duration after the last treatment was “less than 1 year” for 135 patients (34.8%) and “more than 1 year” to “less than 2 years” for 116 patients (29.9%). The subjective health status of the participants was 2.68 ± 0.61 out of 4, and 13.8% responded that they had been diagnosed with depression, musculoskeletal disorders (10.6%), hypertension (8.0%), and diabetes mellitus (5.7%) (Table 2).

### 3.2. Preferences in Forest Healing Program

The responses revealed that 241 participants (62.1%) had heard of forest healing, 28.9% rarely visited the forest, and 28.6% visited the forest once a week. Among the respondents, 27 out of 388 (7.0%) had previously participated in the forest healing program, and 255 (65.7%) were willing to participate.

The survey asked the 255 participants their preferences regarding program frequency and visit duration; the results showed that a regular visit (68.6%) was preferred over a one-time visit. The most preferred program time required was within 2 h (43.9%), followed by within 1.5 h, and within 3 h. Ninety people (35.3%) preferred spring as the season of program operation, eighty-six (33.7%) had no preference, and sixty-six (25.9%) preferred autumn (Table 3).

The most common reasons for not wanting to participate in the forest healing program were lack of time (25.9%), discomfort with strangers (22.4%), lack of friends (13.9%), and poor transportation conditions (13.9%) (Table 4).

### 3.3. Intention to Participate in Forest Healing Program

The binary logistic regression analysis was used to investigate how the intention to participate in the forest healing program related to demographic, clinical, and forest healing program preference factors. Nonsignificant Hosmer–Lemeshow test results validated the fit of the model (*p* = 0.406). The explanatory power of the regression model for the dependent variable was 58.8% (Nagelkerke’s R^2^ = 0.59), and the classification accuracy for the intention to participate in forest healing was 85.3%. For participants with a monthly income of less than USD 1500, the intention to participate in the forest healing program was 5.459 times higher than that of participants with a monthly income of USD 3500 or more. Furthermore, for participants who had ended cancer treatment three to four years ago, the intention to participate in the forest healing program was 3.809 times higher than that of patients who had ended treatment less than a year ago (Table 5).

### 3.4. Purpose of Urban Forest Healing Program

The purpose of participating in the forest healing program was for watching natural landscapes (3.62 ± 0.51), mood change (3.53 ± 0.51), improving immunity (3.52 ± 0.56), improving cardiopulmonary strength (3.51 ± 0.55), and relieving stress (3.50 ± 0.53). There was no statistically significant difference in the purpose of participating in the forest healing program according to the cancer type (Table 6).

## 4. Discussion

This study attempted to understand the perception and preference for forest healing in order to develop a forest healing program for female cancer survivors. The average age of the participants was 45.84 ± 7.96 years, 52.6% were in their 40s, and breast cancer was the most common type of cancer (66.2%). Breast cancer has been reported to have the highest incidence of all cancers in women, not only in Korea but also internationally, and its survival rates are increasing [26,27]. This is consistent with the results of previous studies that surveyed the need for forest healing programs for patients with cancer [28].

Among the participants, 38.1% were diagnosed with other diseases such as depression, musculoskeletal disorders, hypertension, and diabetes. Among hospitalized patients with cancer in Korea, the proportion of those with comorbidities was 74.7%, and circulatory system diseases such as hypertension and hyperlipidemia were the most common types of comorbidities [29]. In a survey by Edwards [30], the prevalence of comorbidities among women aged ≥ 66 years who underwent breast cancer treatment was 32.2%, and the types were hypertension, musculoskeletal disorders, hyperlipidemia, and diabetes. Depression is also a common comorbidity in patients with cancer. According to several previous studies, the prevalence of depression in patients with breast cancer has been reported to be relatively high compared to those with other types of cancer [31,32]. Continuous management is important because patients with cancer and comorbidities have a higher incidence of secondary cancer than those without comorbidities, as well as a negative impact on survival rates and quality of life [33]. The forest healing program has been reported to be effective for depression and hypertension in several previous studies [34,35,36]; therefore, it is thought to be helpful in promoting the health of cancer survivors.

The participants in this study were aware of forest healing, and although their prior participatory experience was low, their willingness to participate was high. These results were similar to studies analyzing the need for forest healing programs for the general public, patients with cancer, and patients with chronic illnesses [23,24,28,32,37]. A high willingness to participate in forest healing is because of an increased interest in health with the improvement in living standards, an increased desire to improve physical and mental health, an interest in the recreational opportunities, and an increased interest in forests which promote psychological and mental health [31,38]. However, cancer survivors have a higher intention to participate in forest healing programs than chronically ill patients or other adults in the general population, possibly owing to their higher interest in seeking methods for improving their health to prevent cancer recurrence. Those who did not want to participate mentioned reasons such as lack of time (25.9%) and inconvenience in meeting other people (22.4%); therefore, it is necessary to utilize easily accessible urban forests and link them with hospitals and local public health centers so that forest healing programs can be part of the integrated support service for cancer survivors. Furthermore, forest healing programs should be especially considered for use by patients with similar diseases.

The 255 participants preferred the regular visit type (68.6%) over the one-time visit type. This is similar to the results of previous studies that targeted urban forest users and patients with cancer [28,39]. In a meta-analysis study on the effects of forest healing on Korean adults [40], the difference in the effect size according to program type was not statistically significant, but the effect size was larger in the regular visit program. Therefore, rather than a one-time program, it is necessary to develop a program that is continuously operating so that patients can regularly participate in forest healing. Moreover, as the reason for not participating in the forest healing program, 13.9% of participants answered that it was inconvenient to travel to the location. Therefore, it is necessary to develop and implement forest healing programs in urban forests that are accessible and can be visited regularly.

The binary logistic regression analysis revealed that the likelihood of participating in the forest treatment program was higher for participants who had finished their cancer treatment three to four years ago compared to those who had finished their treatment less than a year ago. The reason for this perception is that as patients’ health stabilizes and is perceived as having improved to some extent, there is room to carry out health promotion activities. Contrastingly, the higher the participants’ monthly income, the lower the likelihood of participation in the forest treatment program. One possible explanation is that the higher the monthly income, the greater the possibility of implementing various health promotion activities. Participants who frequently visited the forest showed a higher intention to participate in the forest healing program than subjects who rarely visited the forest. This is interpreted as a high intention to participate in the forest healing program as compared to people who have experienced fewer forest visits; those who frequently visit the forest derive a greater sense of satisfaction [41]. Furthermore, the participation intention according to the type of cancer and educational background is considered to be the result of a bias in the number of participants. It is necessary to compare participants with various types of cancer in the future.

The purpose of participation in the forest healing program was confirmed in the following order in this study: appreciation of watching natural landscapes, changing the mood, improving immunity, improving cardiopulmonary strength, and relieving stress. A previous study of urban forest users and adults with diseases reported rest, relaxation, health promotion, and viewing beautiful landscapes [39,42] as the reasons for the aforementioned benefits, similar to the results of this study. These results show that not only cancer survivors but also the general public think that the effects of the forest healing program can be obtained by appreciating natural scenery. According to Ulrich [43], the natural environment has a healing advantage compared to the artificial landscape because it is where human evolution has taken place and humans are instinctively connected with nature. As positive psychological and physiological effects are obtained by watching natural landscapes [44,45], the development of a customized forest healing program for cancer survivors will naturally invite them to the urban forest.

There was no statistically significant difference in the purpose of participating in forest healing programs by cancer type, which is thought to be because of the small number of survivors of various cancer types [28]. In the future, it will be necessary to conduct a more systematic study by classifying it according to various types of cancer.

A limitation of this study was that most of the participants were cancer survivors who had experienced breast cancer, and only women were included. Therefore, future studies that consider sex and carcinoma type are needed. Furthermore, the results revealed that 93% of all respondents had no experience participating in a forest healing program and thus may have only superficially recognized the forest healing program. Consequently, a follow-up study on the preference of those who have experienced forest healing programs is needed. In addition, a survey was conducted online as it was difficult to conduct it face to face owing to the COVID-19 pandemic. This may have excluded those without internet access, and, therefore, as the sample may not be representative of the population, the results may not be generalizable. Additionally, there was no validation of the study’s findings since it was conducted via an online survey. In order to validate the results of future studies, it is necessary to expand the number of participants and carry out repeated studies by conducting face-to-face surveys.

Despite these limitations, this study is meaningful in that it provides the basic data for the development of forest healing programs for cancer survivors through a survey of forest healing awareness and program needs for cancer survivors.

## 5. Conclusions

In this study, the perceptions and preferences of cancer survivors regarding forest healing programs were analyzed. Cancer survivors were highly aware of forest healing, but they had relatively little experience in participating in those programs but were willing to participate. Furthermore, it is necessary to develop a forest healing program that meets the participants’ preferences for seasons as well as the purpose and characteristics of the program and to implement it in the actual field through research on its effectiveness. Additionally, cancer survivors have health problems, such as late-stage side effects, an increased risk of chronic diseases, anxiety, and depression. Therefore, forest healing instructors with the ability to conduct programs that reflect their specificity are required [28,46]. Furthermore, developing a customized forest treatment program for cancer survivors through focus group interviews and qualitative research is needed in the future. Thus, the effects of forest healing programs in urban forests can be evaluated by identifying and catering to the individual needs of cancer survivors.

## Figures and Tables

**Table 1 ijerph-20-01604-t001:** Demographic characteristics (N = 388).

Characteristics	Categories	n (%) or Mean ± SD
Age (years)	≤39 40–49 50–59 ≥60	45.84 ± 7.96 76 (19.6) 204 (52.6) 84 (21.6) 24 (6.2)
Educational level	≤High school University ≥Graduate school	76 (19.6) 270 (69.6) 42 (10.8)
Marriage status	Single Married Divorced/widowed	82 (21.1) 282 (72.7) 24 (6.2)
Caregivers	Spouse and children Parent and sibling Self-care	197 (50.8) 60 (15.5) 131 (33.8)
Employment status	Unemployed Employed	261 (67.3) 127 (32.7)
Monthly household income (USD)	<1500 500–3500 ≥3500	59 (15.2) 165 (42.5) 164 (42.3)
Place of residence	Urban Non-urban	285 (73.5) 103 (26.5)

**Table 2 ijerph-20-01604-t002:** Clinical characteristics (N = 388).

Characteristics	Categories	n (%) or Mean ± SD
Cancer types	Gastrointestinal cancer Thyroid cancer Breast cancer Gynecologic cancer Others ^1^	30 (7.7) 32 (8.2) 257 (66.2) 43 (11.1) 26 (6.7)
End of treatment period (years)	Less than 1 year 1–2 years 2–3 years 3–4 years 4–5 years	135 (34.8) 116 (29.9) 82 (21.1) 42 (10.8) 13 (3.4)
Subjective health	1–4	2.68 ± 0.61
Other disease ^2^	Nothing Hypertension Diabetes mellitus Depression Musculoskeletal disorders	215 (61.8) 28 (8.0) 20 (5.7) 48 (13.8) 37 (10.6)

^1^ Others: Renal cancer, hematologic malignancy, lung cancer, brain tumor, and head and neck cancer. ^2^ Multiple responses.

**Table 3 ijerph-20-01604-t003:** Preference in forest healing program (N = 388).

Items	Categories	n (%) or Mean ± SD
Awareness	Yes No	241 (62.1) 147 (37.9)
Visit frequency	Daily More than once per week More than once per month More than once per quarter Rarely	38 (9.8) 111 (28.6) 82 (21.1) 45 (11.6) 112(28.9)
Experience	Yes No	27 (7.0) 361 (93.0)
Intention to participate	Yes No	255 (65.7) 133 (34.3)
Type of visit (n = 255)	Visit once Visit regularly	80 (31.4) 175 (68.6)
Access time (n = 255)	1 h or less 1 and 1-half h or less 2 h or less 3 h or less 4 h or less 4 h and more	27 (10.6) 54 (21.2) 112 (43.9) 44 (17.3) 13 (5.2) 5 (2.0)
Visiting season (n = 255)	Spring Summer Autumn Winter No preference	90 (35.3) 11 (4.3) 66 (25.9) 2 (0.8) 86 (33.7)

**Table 4 ijerph-20-01604-t004:** Reasons for not participating in the forest healing program (N = 133).

Items	n (%)
Lack of time	52 (25.9)
Uncomfortable with strangers	45 (22.4)
Lack of friends to participate	28 (13.9)
Poor transportation conditions	28 (13.9)
Lack of interest	22 (10.9)
Considered less helpful for health	17 (8.5)
Mobility difficulties	7 (3.5)
Others	2 (1.0)

Note. All items are multiple responses.

**Table 5 ijerph-20-01604-t005:** Result of the binary logistic regression for intention for forest healing program (N = 388).

Variables		B	SE	Wald	*p*	OR	95% CI	
							LL	UL
Age		−0.032	0.026	1.531	0.216	0.969	0.921	1.019
Educational level (ref. ≤ high school)	University	−1.655	0.502	10.857	0.001	0.191	0.071	0.511
	≥Graduate school	0.734	0.784	0.877	0.349	2.084	0.448	9.686
Marriage status (ref. single)	Married	0.566	0.580	0.953	0.329	1.761	0.565	5.482
	Divorced /widowed	1.395	1.195	1.362	0.243	4.034	0.388	41.973
Caregiver (ref. self-care)	Spouse and children	−0.509	0.402	1.605	0.205	0.601	0.274	1.321
	Parent and sibling	0.864	0.615	1.973	0.160	2.373	0.711	7.925
Employment status (ref. unemployed)		0.290	0.360	0.650	0.420	1.337	0.660	2.709
Monthly household income (USD) (ref. ≥ 3500)	<1500	1.697	0.607	7.825	0.005	5.459	1.662	17.931
	1500–3500	1.054	0.381	7.672	0.006	2.869	1.361	6.050
Urban (ref. non-unban)		0.554	0.362	2.348	0.125	1.740	0.857	3.635
Cancer types (ref. gastrointestinal cancer)	Thyroid cancer	0.462	0.881	0.275	0.600	1.588	0.282	8.929
	Breast cancer	3.148	0.688	20.907	<0.001	23.287	6.041	89.767
	Gynecologic cancer	0.973	0.742	1.723	0.189	2.646	0.619	11.320
	Others	1.229	0.850	2.088	0.148	3.417	0.645	18.092
End of treatment period (ref. less than 1 year)	1–2 years	0.007	0.387	0.000	0.986	1.007	0.472	2.150
	2–3 years	0.216	0.451	0.228	0.633	1.241	0.512	3.005
	3–4 years	1.337	0.644	4.306	0.038	3.809	1.077	13.469
	4–5 years	0.326	1.026	0.101	0.750	1.386	0.186	10.355
Subjective health		−0.436	0.295	2.189	0.139	0.647	0.363	1.152
Diagnosed with other disease (ref. no)		0.518	0.360	2.077	0.150	1.679	0.830	3.398
Awareness (ref. no)		0.356	0.340	1.097	0.295	1.427	0.734	2.776
Visit frequency (ref. rarely)	Daily	2.139	0.601	12.678	<0.001	8.495	2.616	27.580
	More than once a week	2.004	0.445	20.309	<0.001	7.421	3.104	17.743
	More than once a month	1.489	0.462	10.394	0.001	4.431	1.793	10.953
	More than once a quarter	0.815	0.523	2.432	0.119	2.260	0.811	6.295
Forest therapy experience (ref. no)		0.271	0.747	0.132	0.716	1.312	0.304	5.666
Constant		−0.968	1.595	0.368	0.544	0.380		
R^2^ (Nagelkerke)		0.588						
Model coefficients χ^2^		215.15 (*p* ≤ 0.001)						
Hosmer and Lemeshow χ^2^		8.28 (*p* = 0.406)						

CI = confidence interval; LL = lower limit; UL = upper limit; ref = reference; OR = odds ratio.

**Table 6 ijerph-20-01604-t006:** Purpose of urban forest healing program (N = 255).

Items ^2^	Overall (1–4)	Gastrointestinal Cancer (n = 10)	Thyroid Cancer (n = 6)	Breast Cancer (n = 210)	Gynecologic Cancer (n = 15)	Others ^1^ (n = 14)	X^2^ (*p*)
1. Mood change	3.53 ± 0.51	136.70	92.92	128.23	153.63	105.86	6.10 (0.192)
2. Relieving stress	3.50 ± 0.53	126.75	126.75	127.36	130.97	135.79	0.27 (0.991)
3. Relieving fatigue	3.30 ± 0.62	128.10	93.75	126.90	143.80	142.18	3.27 (0.514)
4. Pain relief	3.07 ± 0.69	140.30	121.00	124.49	158.70	142.00	4.90 (0.298)
5. Improving immunity	3.52 ± 0.56	123.25	113.75	127.82	135.60	132.07	0.62 (0.961)
6. Sleep promotion	3.41 ± 0.59	123.80	115.67	126.90	143.63	136.00	1.39 (0.846)
7. Cardiopulmonary strength improvement	3.51 ± 0.55	124.50	93.50	126.96	145.33	142.36	3.61 (0.461)
8. Muscle strength improvement	3.41 ± 0.63	134.75	93.00	127.52	138.73	133.93	2.33 (0.675)
9. Watching natural landscapes	3.62 ± 0.51	150.30	154.50	125.25	141.90	127.11	3.56 (0.468)
10. Promoting friendship	3.04 ± 0.75	120.60	105.33	129.97	121.63	120.25	1.33 (0.856)

^1^ Others: Renal cancer, hematologic malignancy, lung cancer, brain tumor, and head and neck cancer. ^2^ Multiple responses.

## Data Availability

The data presented in this study are available upon request from the corresponding author.

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
