# Peer review of "Analysis of Perceptions, Preferences, and Participation Intention of Urban Forest Healing Program among Cancer Survivors"

_ijerph, 2023, doi:10.3390/ijerph20021604_

Round 1
Reviewer 1 Report (Previous Reviewer 3)
In this version, the authors revised several sentences and add some paragraphs to make certain elaboration clear.
One minor suggestion, please use the same terminology for binomial logistic regression and binary logistic regression although in your case they are the same. But try not to use different terminologies for the same term in case some readers might be confused.
Author Response
Please see the attachment.

Reviewer 2 Report (Previous Reviewer 2)
Dear authors, well done!
Author Response
Please see the attachment.

This manuscript is a resubmission of an earlier submission. The following is a list of the peer review reports and author responses from that submission.
Round 1
Reviewer 1 Report
The study appears to be quite preliminary, a survey just to gauge interest of cancer survivors on their preference for forest therapy (FT). I think field data is required to:
a) Let participants select their preference based on actual experience rather than what they “think” that they might like (especially since only 7% had previously participated in FT) and
b) To see if the proposed FT program actually has health benefits and as of now, it is only speculative that the FT based on preference can be beneficial to these cancer survivors.
- What is the assurance that this program structure based on preference correlates with actual health benefits for these cancer survivors?
- How does cancer survivor’s preference differ from regular people? Since they’re no longer undergoing treatment, theoretically they should have the same wants and needs as regular people.
-133 participants stated that they weren’t interested – Are these inhibitors the same or different for regular folks who are not interested in FT?
Author Response
We highly appreciate your advice and interest on our manuscript.
We put the answers in red on your inquiries.
Also, we will upload the answers and location of the body as a table separately.
Once again thanks for taking your time.

Reviewer 2 Report
Patient inclusion:
1. Please, specify why you excluded people more than five years after the end of treatment.
2. What is understood by "insincere responses"?
3. Did the authors intend to calculate the desirable sample size prior to patient enrolment?
Statistical analysis
1. Please, describe the methods used for testing of the normality of data distribution.
Demographic characteristics
1. Please, provide estimates in USD in addition to won.
Overall, why this program was designed for female survivors exclusively?
Author Response

(The authors gave the same response as above.)

Reviewer 3 Report
as attached

Author Response

(The authors gave the same response as above.)

Round 2
Reviewer 1 Report
The manuscript has undergone significant improvement. I think the conclusion can still be improved - Please add limitations of the study (there was no validation for the study's findings since it was done via an online survey). And also add recommendations for future research.
Also, Line 104 (Each question was kept as a "All questions were mandatory."), I am not sure what is meant by this, it may need to be better worded out.
Reviewer 3 Report
as attached
